# Increased Collagen I/Collagen III Ratio Is Associated with Hemorrhage in Brain Arteriovenous Malformations in Human and Mouse

**DOI:** 10.3390/cells13010092

**Published:** 2024-01-01

**Authors:** Zahra Shabani, Joana Schuerger, Xiaonan Zhu, Chaoliang Tang, Li Ma, Alka Yadav, Rich Liang, Kelly Press, Shantel Weinsheimer, Annika Schmidt, Calvin Wang, Abinav Sekhar, Jeffrey Nelson, Helen Kim, Hua Su

**Affiliations:** 1Center for Cerebrovascular Research, University of California, San Francisco, CA 94143, USA; zahra.shabaninabikandi@ucsf.edu (Z.S.); joana.schuerger@rwth-aachen.de (J.S.); zxn@whu.edu.cn (X.Z.); chaolt@ustc.edu.cn (C.T.); lima.neuro@gmail.com (L.M.); alka.yadav@ucsf.edu (A.Y.); liangr71@gmail.com (R.L.); mpress@gmail.com (K.P.); shantel.weinsheimer@ucsf.edu (S.W.); annika.schmidt1@rwth-aachen.de (A.S.); calvin.wang@ucsf.edu (C.W.); abi.sekhar@ucsf.edu (A.S.); jeffrey.nelson@ucsf.edu (J.N.); helen.kim2@ucsf.edu (H.K.); 2Department of Anesthesia and Perioperative Care, University of California, San Francisco, CA 94143, USA

**Keywords:** brain arteriovenous malformation, collagen I, collagen III, microhemorrhage, depletion of microglia, gluycoprotein nonmetastatic B

## Abstract

**Background:** The increase in the collagen I (COL I)/COL III ratio enhances vessel wall stiffness and renders vessels less resistant to blood flow and pressure changes. Activated microglia enhance inflammation-induced fibrosis. **Hypotheses:** The COL I/COL III ratio in human and mouse brain arteriovenous malformations (bAVMs) is associated with bAVM hemorrhage, and the depletion of microglia decreases the COL I/COL III ratio and hemorrhage. **Method:** COL I, COL III, and hemorrhages were analyzed in 12 human bAVMs and 6 control brains, and mouse bAVMs induced in three mouse lines with activin receptor-like kinase 1 (*n* = 7) or endoglin (*n* = 7) deleted in the endothelial cells or brain focally (*n* = 5). The controls for the mouse study were no-gene-deleted litter mates. Mouse bAVMs were used to test the relationships between the Col I/Col III ratio and hemorrhage and whether the transient depletion of microglia reduces the Col I/Col III ratio and hemorrhage. **Results:** The COL I/COL III ratio was higher in the human and mouse bAVMs than in controls. The microhemorrhage in mouse bAVMs was positively correlated with the Col I/Col III ratio. Transient depletion of microglia reduced the Col I/Col III ratio and microhemorrhage. **Conclusions:** The COL I/COL III ratio in the bAVMs was associated with bAVM hemorrhage. The depletion of microglia reduced the bAVM Col I/Col III ratio and hemorrhage.

## 1. Introduction

Intracerebral hemorrhage (ICH) constitutes around 10–15% of strokes and causes high early mortality in patients [1]. Brain arteriovenous malformations (bAVMs) are one of the most frequent sources of non-traumatic ICH. About 50% of bAVM cases present initially with ICH [1,2]. A brain AVM is a tangle of dysplastic vessels that shunt blood directly from the arteries to veins [3]. The progression of bAVMs to ICH remains incompletely understood.

The extracellular matrix (ECM) is a main constituent of the cellular microenvironment and generates an intricate three-dimensional network [4]. The vascular ECM primarily consists of the basement membrane and the interstitial ECM. The interstitial ECM is composed of elastic fibers and fibrillar collagens and is located in the media surrounding the vascular smooth muscle cells (SMCs) of the large vessel walls, which maintains normal vessel wall stiffness and elasticity to mechanically control shear stress and pressure [5].

Elastin and collagens are major ECM elements in the arteries. Seventeen collagen types have been recognized in the mouse aorta. The collagen fibers are disorganized and interrupted in the internal elastic lamina of bAVM vessels [6]. A close relationship between collagen structure and vascular mechanical load was shown [6]. In addition, the collagen catabolic process and genes encoding different subtypes of collagen are highly enriched in ruptured bAVMs [7].

Collagen I (COL I) and COL III are the most abundant collagens in the vascular wall and regulate vascular homeostasis and remodeling [8]. COL I and III fibers are connected, entangled, and packed to make a firm structure [9]. COL I provides resistance to stretching, whereas COL III forms an elastic network that offers resilience and structural maintenance of the vessel wall. Several studies emphasized the contribution of the collagen system to the pathophysiology of sporadic bAVMs [10,11]. Single-cell RNA sequence (RNAseq) data show that perivascular fibroblasts and fibromyocytes, which are the major cells that produce COLs, are strongly associated with the development of sporadic bAVMs in humans and are found in proximity to the peri-lesional macrophages [12]. In addition, the contents of COL I and COL III in ruptured AVMs were higher than those of non-ruptured AVMs [9]. Nevertheless, the mechanisms underlying these changes have not been studied. 

It was found that perivascular macrophages are involved in collagen production around cerebral small vessels [13] and microglia accumulation is associated with fibroblast proliferation and subsequent collagen production. Gluycoprotein nonmetastatic B (GPNMB)^+^ monocytes were also identified in human bAVMs [12], which contribute to SMC depletion [12]. These cells may also be involved in collagen synthesis in part through activating fibroblasts [3,14,15]. Further, it was shown that the crosstalk of platelets with macrophages and fibroblasts aggravates inflammation and aortic wall stiffening [16]. Thus, microglia and monocytes may play roles in the activation of fibroblasts in bAVMs and contribute to collagen synthesis.

In this study, we aimed to address the following questions: (1) whether an increase in the COL I/COL III ratio plays a role in bAVM hemorrhage and (2) whether a reduction in bAVM inflammation via the transient depletion of microglia reduces the COL I/COL III ratio and bAVM hemorrhage. We first examined the COL I and COL III levels in surgically resected human bAVM tissues. Since COL I provides resistance to stretching, whereas COL III forms an elastic network that offers resilience of the vessel wall, an increase in the COL I/COL III ratio will increase the vessel stiffness and render the vessel less resistant to changes in flow and pressure. Therefore, we also analyzed the COL I/COL III ratio to determine whether there is any correlation between the COL I/COL III ratio and bAVM hemorrhage. Mouse bAVM models were used to test possible mechanisms underlying the observed correlation of the COL I and COL III levels and COL I/COL III ratio with hemorrhage in bAVMs and to test whether the transient depletion of microglia reduces bAVM hemorrhage. 

## 2. Materials and Methods

### 2.1. Ethics Statement

Human brain tissues and clinical data were obtained from the University of California San Francisco (UCSF) with protocols approved by the Institutional Review Board and Ethics Committee (IRB: 10-02012). All tissues were acquired from patients undergoing neurosurgical operations and written informed consent was obtained before the procedure permitting the collection of tissue specimens for research. For nonvascular lesion controls, temporal lobe specimens were similarly acquired from subjects undergoing anterior temporal lobectomy for medically refractory epilepsy.

In this study, we took advantage of a large cohort of patients with bAVM in place as part of the UCSF Brain AVM Project, which was led by co-author Dr. Helen Kim (R01 NS034949). Dr. Kim’s team discussed the study with all patients to obtain consent to participate in our research studies, which allowed for tissue to be harvested, banked, and analyzed for medical research. For this study, we used de-identified tissue specimens from 12 (7 unruptured and 5 ruptured) sporadic bAVM subjects and 3 HHT patients undergoing surgery, with appropriate consent. Nonvascular lesion controls and temporal lobe specimens were selected from subjects that were matched to bAVM cases using the demographic distribution from enrolled cases at UCSF. 

All bAVM patients seen for evaluation or treatment at UCSF were considered. There was no internal bias in enrollment and no proposed exclusion of any sex, age, or racial/ethnic group. UCSF Medical Centers at Parnassus and Mission Bay are large tertiary referral hospitals serving the San Francisco Bay area, which is well known for its rich diversity of cultures. 

Samples that had bAVM tissues or temporal lobe tissues without damage/sclerosis were banked and used in this study. 

All animal experimental protocols were approved by the Institutional Animal Care and Use Committee (IACUC) of the UCSF. The staff of the IACUC of UCSF Animal Core Facility provided animal husbandry according to the guidance of certified animal technologists. Veterinary care was offered by IACUC faculty members and veterinary inhabitants located on the San Francisco General Hospital campus. All mice were maintained in a pathogen-free area in 421 × 316 cm^2^ cages and were kept on a 12 h light and dark cycle with free access to food and water. All animal experiments were done by co-authors who were authorized by the IACUC of the UCSF to perform experiments on mice (please also see Institutional Review Board Statement section).

### 2.2. Human Specimens

Diagnoses of human sporadic bAVMs (12 total, with 7 unruptured and 5 ruptured) were confirmed via preoperative angiography, and fresh tissues were acquired as part of planned surgical resection. We also collected bAVMs from 3 HHT patients. Temporal lobe specimens similarly acquired from 6 subjects undergoing anterior temporal lobectomy for medically refractory epilepsy were used as the controls. All specimens were fresh frozen in OCT at the time of sample collection and analyzed at the same time in this study. Samples were sectioned into 20 μm thick sections using a Leica RM2155 microtome (Leica Microsystems, Wetzlar, Germany) [17]. The demographic information of individuals in the sample group is shown in Appendix A.

### 2.3. Animals

We used 8-to-10-week-old mice in this study. Both male and female mice were used. For the induction of bAVMs in the mice, the following genetically modified mouse lines were used: (1) *Pdgfb*icreER;*Eng*^f/f^ (*n* = 7), (2) *Pdgfb*icreER;*Alk1*^f/f^ (*n* = 7), and (3) *Alk1*^f/f^ (*n* = 5). Litter mates without gene deletion were used as the control (*n* = 7). The *Pdgfb*icreER;*Eng*^f/f^ and *Pdgfb*icreER;*Alk1*^f/f^ have platelet-derived growth factor (pdgf) promoter-driven estrogen-inducible cre recombinase expression in the endothelial cells (ECs) [18] and have endoglin (*Eng*) gene exons 5-6 floxed [19] or activin receptor line kinase 1 (*Alk1,* also known as *Acvrl1*) gene exons 4-6 floxed [20]. *Alk1*^f/f^ mice have *Alk1* gene exons 4-6 floxed [20]. Using these mouse lines, we tested the effects of two bAVM-causative genes, namely, *Eng* and *Alk1*, and two gene-deletion sides in ECs or locally in the brains. The inclusion criteria were (1) 8 to 10 weeks of age and (2) no wound infection or neuronal deficits after receiving the model induction procedures (please see Section 2.5). The exclusion criteria were (1) mice that were younger than 8 weeks or older than 10 weeks or pregnant; (2) wound infection or neuronal deficits after receiving the model induction procedures (please see Section 2.5); and (3) brain samples were damaged during sample collection, sectioning, or staining (no sample damage occurred during this study).

### 2.4. Brain AVM Model Induction and Viral Vector Delivery in Mice

Brain AVMs were induced in *Pdgfb*icreER;*Alk1*^f/f^ mice (*n* = 7) through the intra-brain injection of an adeno-associated viral vector expressing vascular endothelial growth factor (AAV-VEGF, 2 × 10^9^ genome copies (vgs)) to induce brain focal angiogenesis and intra-peritoneal (i.p.) injection of tamoxifen (TM, 1.25 mg/kg of body weight) 14 days later to delete the *Alk1* gene in the ECs (Figure 1A) [21,22] since angiogenic stimulation is necessary for the induction of bAVM in adult mice [21,23,24,25,26] and TM treatment is needed to activate icre in the *Pdgfb*icreER transgene. Brain AVMs in *Pdgfb*icreER;*Eng*^f/f^ mice (*n* = 7) were induced through intra-brain injection of AAV-VEGF (2 × 10^9^ vgs) on day 1 and i.p. injection of TM (2.5 mg/kg) for three consecutive days starting on day 1 to delete *Eng* gene in ECs (Figure 1B) [25]. Control *Pdgfb*icreER;*Alk1*^f/f^ mice (*n* = 7) and *Pdgfb*icreER;*Eng*^f/f^ mice (*n* = 7) received an intra-brain injection of AAV-VEGF and i.p. corn oil injection. Brain AVM in *Alk1*^f/f^ (*n* = 5) mice was induced via co-injection of an adenoviral vector expressing Cre recombinase (Ad-Cre, 2 × 10^7^ plaque-forming units) to delete the *Alk1* gene in the brain focally and AAV-VEGF (2 × 10^9^ vgs) to induce brain angiogenesis (Figure 1C) [23]. Control mice for the *Alk1*^f/f^ model (*n* = 5) received intra-brain injections of Ad-GFP and AAV-VEGF. Brain samples were collected from all groups 8 weeks after the model injection for analyses (Figure 1).

For the injection of viral vectors into mouse brain, the mice were anesthetized with 4% isoflurane inhalation and were placed in a stereotactic apparatus with a mouth holder (David Kopf Instruments, Tujunga, CA, USA). A burr hole was drilled in the pericranium 2 mm lateral and 1 mm posterior to the bregma. A 10 µL Hamilton syringe was inserted into the right basal ganglia 3 mm beneath the brain surface. Two microliters of viral suspension were slowly injected at a rate of 0.2 µL per minute. The needle was withdrawn after 10 min, and the wound was closed with a 4-0 suture.

### 2.5. Administration of CSF1R Inhibitor (PLX5622)

PLX5622 (180 mg/kg of body weight/day, Plexikon Biotech Company, South San Francisco, CA, USA) was incorporated in chow and orally administered for 7 consecutive days starting at 8 weeks after the bAVM model induction in *Alk1*^f/f^ mice (*n* = 5) based on the instructions provided by Plexikon Biotech Company. The placebo chow was administrated in the same manner as the control group (Figure 1D).

### 2.6. Immunofluorescence Staining of Human and Mouse Sections

For human samples, two sections of each bAVM and control brain tissue underwent immunofluorescence staining using the protocol we established in our laboratory [17]. Sections were co-stained with antibodies specific to human COL I and CD31 (EC marker) or COL III and CD31 using rabbit anti-COL I (1:250, NBP1-30054, Novus Bio-Tecne, Centennial, CO), goat anti Col III (1:250, 1330-01, Southern Biotec, Birmingham, AL, USA), and mouse anti-CD31 (1:100, 14-0319-82, eBioscience, San Diego, CA, USA). Sections were incubated with primary antibodies at 4 °C overnight. After washing with PBS, donkey anti-mouse antibody conjugated with Alex Flour 594 (Thermo Fisher Scientific, Cat#A21203, 1:200), donkey anti-goat antibody conjugated with Alexa Fluor 488 (Thermo Fisher Scientific, Cat#A1105, 1:300), and donkey anti-rabbit antibody conjugated with Alexa Fluor 488 (Thermo Fisher Scientific, Cat#A32790, 1:300) were used as secondary antibodies to visualize positive stains. Vectashield anti-fade mounting medium containing 4′,6-diamidino-2-phenylindole (DAPI) (Vector Laboratories, Cat# H-1200, Burlingame, CA, USA) was used to stain the cell nuclei and mount the slide. Sections were examined and imaged using a Keyence fluorescence microscopy under a 20× objective lens (Model BZ-9000, Keyence Corporation of America, Itasca, IL, USA). A total of 10 images were taken from each brain sample, with five from each section, for analyses.

To study the mouse samples, the mice were anesthetized with 4% isoflurane inhalation, and their brains were collected, frozen in dry ice, and cut into 20 μm thick coronal sections with a Leica RM2155 Microtome (Leica Microsystems, Wetzlar, Germany) [21,23,24,25,26]. Two sections per brain adjacent to the injection site were selected and co-stained with Col I and CD31 or Col III and CD31 using mouse anti-Col I (1:250, NB600-450, Novus Biologicals, Littleton, CO), goat anti-Col III (1:250, 1330-01, Southern Biotechnoloty Associates Inc., Birmingham, Ala), and rat anti-CD31 (1:100, SC-18916, Santa Cruz, CA, USA) antibodies. For the Col I stain, sections were blocked for 1 h in the M.O.M. kit blocker (one drop/5 mL, MKB-2213-1, Vector Laboratory, Newark, CA, USA). After incubating at 4 °C overnight with primary antibodies, sections were washed with PBS and were incubated with donkey anti-rat antibody conjugated with Alexa Fluor 488 (Thermo Fisher Scientific, Waltham, MA, Cat #A-21208, 1:100), donkey anti-mouse antibody conjugated with Alexa Fluor 594 (1:300), and donkey anti-goat antibody conjugated with Alexa Fluor 594 (Thermo Fisher Scientific, Cat #A-11058, 1:300) for 1.5 h at room temperature. CD68^+^ and GPNMG^+^ cells were stained on two sections per mouse brain adjacent to the injection site with a rat anti-mouse CD68 antibody (1:250 MCA1957, Bio-Rad, Hercules, CA, USA) and rabbit anti-GPNMB (AB188222, 1:200 abcam, Cambridge, UK), as described above. The sections were then incubated with donkey anti-rabbit antibody conjugated with Alexa Fluor 488 (Thermo Fisher Scientific, Cat #A-21206, 1:400) for 1.5 h at room temperature. After being incubated with the secondary antibodies, all sections were mounted with Vectashield anti-fade mounting medium DAPI (Vector Laboratories, Birlingame, CA, Cat# H-1200). Mouse brain sections were examined and imaged using a Keyence fluorescence microscopy under a 20× objective lens (Model BZ-9000, Keyence Corporation, Osaka, Japan). A total of six images were taken from each brain sample, with three from each brain section (to the right, to the left and below the injection site).

The images were coded by a researcher who did not participate in the quantification. All quantifications were performed by at least two researchers who were blinded to the group assignment. COL I, COL III, and CD31 stained areas were quantified using NIH image J 1.63 software. The number of CD68^+^ cells and GPNMB^+^ cells were counted manually [21,23,24,25,26].

### 2.7. Prussian Blue Staining

Two sections per brain adjacent to the injection site were used for detecting iron deposition using an Iron Stain Kit (Sigma-Aldrich, St. Louis, MO, USA). Slides were incubated in a freshly prepared working iron stain solution for 15 min, washed in distilled water, and then counterstained with pararosaniline solution for 3 min. Data are presented as Prussian blue-positive area (mm^2^).

### 2.8. RNA-Seq Analysis

Brain AVM lesions (*n* = 3) and brain angiogenic regions (controls, *n* = 3) were identified via needle injection holes on the brain surface and the depth of needle tips (3 mm beneath the brain’s surface; please see Section 2.4). One mm^3^ tissue around the needle tip was collected from each brain for RNA isolation. The total RNA isolated from these tissues was sent to Novogene Co (Davis, CA, USA) for sequencing using the company’s standard protocol (Appendix A). The outcome data were also analyzed by Novogen Co.

### 2.9. Statistical Analysis

For quantification of the dependent variables COL I and COL III levels, CD68^+^ cells, GPNMB^+^ cells, and Prussian blue-positive areas, the section numbers were scrambled. The quantification was done independently by at least two researchers who were blinded to the treatment groups. A histogram was used to test whether the data were normally distributed. Data are represented as mean  ±  SD. The differences between the two independent variables (groups shown in figures) were analyzed using two-sample *t*-tests. A *p*-value < 0.05 was considered to be significant. We applied a log transformation (log Y) to the Col I and Col III levels before performing analysis because the data were not normally distributed. Sample sizes were determined via a power calculation using data obtained by pilot studies and indicated in Section 2.3 and Section 2.5 and the figures.

## 3. Results

### 3.1. Human Ruptured bAVMs Had Higher COL I/COL III Ratios Than Unruptured bAVMs

We found that the levels of COL I and COL III were increased in both the ruptured (2.49 ± 0.12 for COL I, 2.22 ± 0.12 for COL III, *ps* < 0.001) and unruptured (2.43 ± 0.06 for COL I, 2.26 ± 0.06 for COL III, *ps* < 0.001) sporadic bAVMs compared with the controls (1.87 ± 0.03 for COL I, 1.79 ± 0.06 for COL III; Figure 2A–F). The COL I/COL III ratio was higher in the ruptured bAVMs compared with the unruptured bAVMs (1.90 ± 0.27 for ruptured bAVMs vs. 1.46 ± 0.15 for unruptured bAVMs, *p* = 0.005; Figure 2G). These data revealed that the increase in the COL I/COL III ratio was associated with bAVM rupture.

The COL I and COL III levels in human HHT bAVMs were also higher than in the controls (COL I: *p* < 0.001 and COL III: *p* < 0.001; Appendix A). In addition, the HHT AVMs displayed a higher COL I/COL III ratio than the controls (*p* = 0.034). Our results indicate that the COL I and COL III levels, as well as the COL I/COL III ratio, were also increased in the HHT bAVMs.

### 3.2. Mouse bAVMs Had Higher Col I/Col III Ratios, Which Were Associated with Microhemorrhage

We found that bAVMs in the *Pdgfb*icreER;*Alk1*^f/f^ and *Pdgfb*icreER;*Eng*^f/f^ mice exhibited higher levels of Col I and Col III compared with brain angiogenic regions in the corn-oil-treated control mice. In the *Pdgfb*icreER;*Alk1*^f/f^ mice, the Col I level in the bAVMs was 1.88 ± 0.11, while in the brain angiogenic region of the corn-oil-treated mice, it was 1.22 ± 0.09 (*p* < 0.001). The Col III level in the bAVMs of the *Pdgfb*icreER;*Alk1*^f/f^ mice was 1.72 ± 0.20, while in the brain angiogenic region of the corn-oil-treated mice, it was 1.35 ± 0.14 *(p* = 0.002). The Col I/Col III ratio in the *Pdgfb*icreER;*Alk1*^f/f^ bAVMs (1.72 ± 0.20) was also higher than that in brain angiogenic region of the corn-oil-treated mice (1.35 ± 0.34, *p* = 0.006; Figure 3A,B,D,E,G).

In *Pdgfb*icreER;*Eng*^f/f^ mice, the Col I level in the bAVMs was 1.60 ± 0.14, while in the brain angiogenic region of corn-oil-treated mice, it was 1.22 ± 0.09 (*p* < 0.001). The Col III level in the bAVMs of *Pdgfb*icreER;*Eng*^f/f^ mice was 1.53 ± 0.10, while in the brain angiogenic region of corn-oil-treated mice, it was 1.35 ± 0.14 *(p* = 0.02). The Col I/Col III ratio in the *Pdgfb*icreER;*Eng*^f/f^ bAVMs (2.08 ± 0.10) was also higher than in the brain angiogenic region of corn-oil-treated mice (1.87 ± 0.17, *p* = 0.02; Figure 3A,C,D,H).

Microhemorrhages in the bAVMs and control brain angiogenic regions were analyzed using Prussian blue staining. We found more iron depositions in the bAVMs of *Pdgfb*icreER;*Alk1*^f/f^ (*p* = 0.015) and *Pdgfb*icreER;*Eng*^f/f^ (*p* = 0.016) compared with the brain angiogenic regions of the corn-oil-treated mice (Figure 4), indicating the presence of microhemorrhage in the bAVMs. The Prussian blue-positive areas were positively correlated with the Col I/Col III ratio (*p* =  0.007, r = 0.52, r^2 =^ 0.27; Figure 4D). Collectively, these data suggest that a higher Col I/Col III ratio was associated with greater microhemorrhage in mouse bAVMs.

### 3.3. Transient Depletion of Central Nervous System (CNS) Macrophages/Microglia Reduced Col I/Col III Ratio and Microhemorrhage in bAVMs

To test whether there was a reduction in inflammation in the mouse bAVMs, such as increased inflammatory cells [27], via a transient depletion of CNS macrophages/microglia influences Col I and Col III levels and the Col I/Col III ratio in the bAVMs, *Alk1*^f/f^ mice were treated with PLX5622 8 weeks after the bAVM model induction for 7 days. The PLX5622 treatment reduced Col I level (*p* = 0.003 versus the controls). But did not affect the Col III levels (*p* = 0.33, Figure 5A–D). The Col I/Col III ratio (*p =* 0.04) and microhemorrhage in the PLX-treated bAVMs were lower than in the controls (*p* < 0/001, Figure 5E and Figure 6E,F). The PLX5622 treatment reduced the CD68^+^ cells in the bAVMs (*p* = 0.048 versus the controls, Figure 6A,B). These data reveal that the transient depletion of CNS macrophages/microglia decreased the Col I/Col III ratio and microhemorrhage in the bAVMs.

Since it was shown that GPNMB^+^ macrophages may contribute to collagen synthesis in part through activating fibroblasts [3,14,15], we analyzed whether PLX5622 treatment affects GPNMB^+^ macrophages. We found that PLX5622 treatment not only reduced the CD68^+^ microglia/macrophages (*p* = 0.048, Figure 6A,B) but also reduced the number of GPNMB^+^ macrophages in the bAVMs (*p* =  0.042, Figure 6C,D). The numbers of CD68^+^ and GPNMB^+^ were positively corrected with the Col I/Col III ratio and microhemorrhages in the bAVMs (Figure 6E–H). Collectively, these data suggest that the reduction in CD68^+^ and GPNMB^+^ cells reduced the Col I/Col III ratio and microhemorrhages in the bAVMs.

### 3.4. Fibroblast- and Inflammatory-Related Genes Were Upregulated in Mouse bAVMs

We identified 2143 genes that were upregulated and 1830 genes that were downregulated in the bAVMs compared with the controls. In addition to the upregulation of angiogenesis (*padj* = 2.07 × 10^−17^), vasculogenesis (*padj* = 9.6 × 10^−5^), and acute inflammatory response (*padj* = 0.042) signaling, collagen binding signaling was upregulated (*padj* = 0.0001) in the bAVMs analyzed using Gene Ontology (GO) enrichment analyses. Differential analysis showed the upregulation of Col1a1 (*padj* = 0.035) and Col3a1 (*padj* = 0.032) gene expressions in the bAVMs compared with the controls. Several inflammatory-related genes were also upregulated in the bAVMs, such as CD68 (*padj* = 0.004), Icam2 (*padj* = 0.0003), and Csf1 (*padj* = 0.002). In addition, the expressions of the fibroblast genes Vim (*padj* = 0.019), CD34 (*padj* = 0.02), CD248 (*padj* = 1.8 × 10^−5^), Csf2rb (*padj* = 0.049), and Mpeg1 (*padj* = 1.7 × 10^−5^) were increased in the bAVMs (Table 1). Interestingly, the Timp2 (*padj* = 0.01) and Timp3 (*padj* = 0.04) expressions were also increased in the bAVMs (Table 1), which could reduce collagen degradation mediated by MMPs.

## 4. Discussion

In this study, we demonstrated that COL I and COL III levels, as well as the ratio of COL I/COL III, were increased in the human and mouse bAVMs. The ruptured sporadic bAVMs had higher COL I/COL III ratios compared with the unruptured bAVMs. The results obtained from mouse bAVM models were consistent with those obtained from human specimens. In the mice, the Col I/Col III ratio was positively correlated with the degree of bAVM hemorrhage. Furthermore, our data illustrate a positive correlation between an increased Col I/Col III ratio and the degree of microhemorrhage in mouse bAVM models. In addition, transient depletion of CNS macrophages/microglia reduced the Col I/Col III ratios and microhemorrhage in the mouse bAVMs.

Our results are consistent with previous findings. It was reported in a study with 43 patients that severe EC damage, fewer SMCs in the media, and hyperplasic COL I and COL III content are present in ruptured bAVMs [9]. Guo et al. reported that collagen fibers in the internal elastic lamina of blood vessels in the human bAVM nidus are disorganized and interrupted. They also noticed an increase in COL I and a decrease in COL III levels in human bAVMs compared with control brain samples [2]. Fu et al. showed that vascular fibrosis was a common pathological manifestation in bAVMs with or without microhemorrhage. However, collagen deposition was notably higher in bAVMs with microhemorrhage than those without [28]. Their data confirmed the correlation of collagen with bAVM microhemorrhage. However, they did not distinguish different types of collagens in their study. The contribution of COL I and COL III imbalance to bAVM hemorrhage was also reported by Neu et al. [9].

COL I and COL III are interstitial or fibrous collagen types, which are connected, entangled, and packed to generate a steady structure [29], wrapping the blood vessels to form an almost continuous layer [30]. COL I provides strength and resistance to the stretch and deformation of vessels, while Col III presents elasticity and resilience [31]. COL III fibers are thinner and more elastic, which afford less tensile strength than COL I, thus providing more resilience, tissue distensibility, and structural support [32]. The combination and organization of these collagen subtypes maintain the function, integrity, and strength of blood vessels. An increase in the COL I/COL III ratio in a bAVM may represent impairment of the ECM, leading to structural instability of the vasculature in a bAVM and vessel wall weakness [33].

The balance of the Col I/Col III ratio is maintained by multiple mechanisms. The levels of COL I and COL III depend not only on their production but also on their degradation [29]. Hence, a higher COL I/COL III ratio in bAVMs could be caused by the overproduction of COL I or faster degradation of COL III. It was shown that the activation of vascular TGF-β1 and its downstream signaling effector SMAD increases the synthesis of ECM proteins, such as COL I, in fibroblasts. TGF-β also reduces collagenase production and inhibits COL degradation [34,35]. The fibroblast cells are predominant resident cells in the adventitial layer and are responsible for depositing abundant collagen fibrils around vessels. In addition, it was suggested that pericytes can express Col Ia1 and Col IIIa1 [35,36]. Our RNA-seq data show that the expression of Col1a1 and Col IIIa1 were increased in the bAVMs, suggesting that Col I and Col III productions were increased in the bAVMs.

On the other hand, endopeptidases, such as matrix metallopeptidases (MMPs), can degrade ECM components, including collagens. Increased MMP activity can reduce vessel wall stability and integrity and blood–brain barrier integrity, and thus, increase the risk of hemorrhage [37,38]. MMP-1 is synthesized by fibroblasts and shows equal effects for COL I and COL III degradation, while MMP-8, which is synthesized by neutrophils, has a higher affinity for COL III. The MMP activities can be inhibited by tissue inhibitors of metalloproteinases-1 (TIMP-1) to TIMP-4 [39,40]. Therefore, the altered activities of MMPs and TIMPs may contribute to the changes in COL I and COL III ratios in a bAVM [41,42]. Indeed, we found using RNAseq that the expressions of Timp2 and 3 were increased in the bAVMs of *Pdgfb*icre;ER;*Eng*^f/f^ mice, which could be one of the mechanisms responsible for the increase in the Col I and Col III levels in the bAVMs.

We showed that the transient depletion of microglia/macrophages by PLX5622 treatment reduced the Col I/Col III ratio, microhemorrhage, CD68^+^, and GPNMB^+^ cells in the bAVMs of *Alk1^f/f^* mice. The Col I/Col III ratio and the degree of microhemorrhage were positively correlated with CD68^+^ and GPNMB^+^, suggesting that inflammation played a role in the increased Col I/Col III ratio in the bAVMs. Microglia, which are the primary resident immune cells of the CNS, are key elements of neuroinflammation. Enhanced accumulation of microglia was observed in bAVMs in both human and mouse models [43,44,45]. It was shown that macrophage infiltration was significantly higher in bAVMs with hemorrhage than those without and the normal controls [46]. It was shown that neuroinflammation is present on day 1 post-ICH, with changes in microglial size and morphology, which can be persistent up to day 12 [47]. Nevertheless, the link between microglia activation and hemorrhage is unclear.

A handful of studies indicated the association between microglia activation and collagen level change [2,48]. Interestingly, a recent study demonstrated that Col I production is enhanced around cerebral small vessels in rats with prolonged hypertension and *Col1a1* is expressed by perivascular macrophages, supporting the concept that perivascular macrophages (PVMs) contribute to collagen production and vascular fibrosis [13]. However, the correlations between microglia/macrophages with Col I/Col III ratios and hemorrhage have been poorly explored.

GPNMB is an endogenous type 1 transmembrane glycoprotein [49]. In the CNS, GPNMB is expressed in monocytes, where it plays an important role in the regulation of inflammatory responses [50]. Winkler et al. revealed that the interaction between vascular and immune cells, such as GPNMB^+^ monocytes, can induce pathological changes associated with brain hemorrhage. They showed that GPNMB^+^ monocytes contribute to SMC depletion and are associated with bAVM rupture and brain hemorrhage [12]. It was reported that GPNMB may activate resident normal fibroblasts [3]. Thus, microglia and monocytes may play roles in the activation of fibroblasts in bAVMs, resulting in increased collagen production and bAVM hemorrhage.

Our RNA-seq results show that *Cd68*, *Csf1*, and csf1r were upregulated in the mouse bAVMs. Csf-1 is a cytokine required for the differentiation of monocyte lineage cells. It was shown that csf1r expression was induced during macrophage differentiation in mice. Moreover, *Gpnmb* RNA levels are regulated by Csf-1 during the process of macrophage differentiation [51]. According to these results, it is quite likely that Gpnmb^+^ cells and activated macrophages are associated with changes in Col I and Col III levels via the activation of residing fibroblasts and increased production of collagen fibers. Consistent with our data, a recent study showed that the collagen catabolic process and genes encoding different subtypes of collagen, as well as several inflammatory terms, like the innate immune response and chemotaxis, were highly enriched in the ruptured bAVMs, which indicate the important role of inflammatory response in collagen production [7].

In addition to COL I and III, COL IV was also implicated in bAVM pathogenesis. In capillary malformation-AVM with Ephrin type-B receptor 4 (*Ephb4*) or Ras p21 protein activator 1 (*Rasa1*) mutation, Col IV accumulates in the EC endoplasmic reticulum, leading to EC apoptotic death [52,53]. An age-dependent increase in COL4A2 was also reported in bAVMs, suggesting a potential role of COL4A2 in bAVM pathophysiology [54]. We will analyze COL IV and its relationships with COL I and III and inflammatory cells in the future.

## 5. Conclusions

Our results show that an increased COL I/COL III ratio is associated with an increase in bAVM microhemorrhage and rupture, probably via increased activated macrophages/macroglia and GPNMB^+^ cells. Inhibition of inflammation through the transient depletion of CNS macrophages/microglia reduced Col I/Col III ratios and microhemorrhage in the mouse bAVMs. Therefore, the inhibition of inflammation can be a target for the development of therapies to reduce bAVM hemorrhage.

## Figures and Tables

**Figure 1 cells-13-00092-f001:**
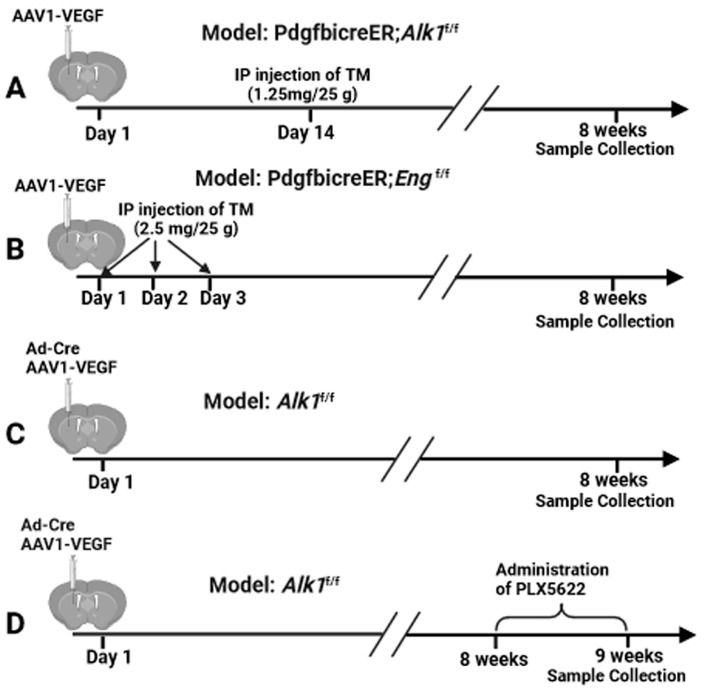
Study design. (**A**–**C**) Induction of bAVMs in *Pdgfb*icreER;*Alk1*^f/f^, *Pdgfb*icreER;*Eng*^f/f^, and *Alk1*^f/f^ mice. (**D**) PLX5622 treatment.

**Figure 2 cells-13-00092-f002:**
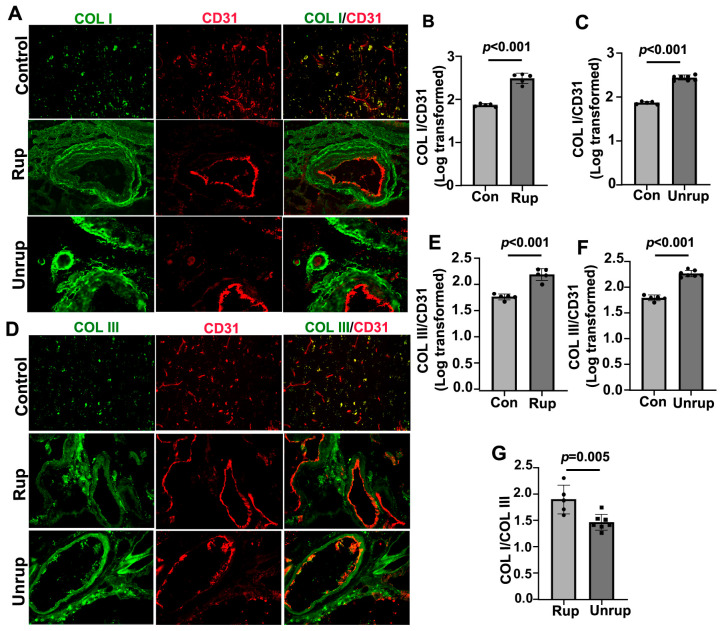
COL I/COL III ratio was higher in ruptured than unruptured bAVMs. (**A**,**D**) Representative images of COL I (green) and COL III (green) antibody-stained sections. ECs (red) were stained using an anti-CD31 antibody. Scale bar: 50 µm. (**B**,**C**,**E**,**F**) Quantifications of COL I and COL III levels on vessels. (**G**) Quantification of COL I/COL III ratio. Con: controls; Rup: ruptured bAVMs; Unrup: unruptured bAVMs. *n* = 6 for control, *n* = 5 for ruptured bAVMs, and *n* = 7 for unruptured bAVMs.

**Figure 3 cells-13-00092-f003:**
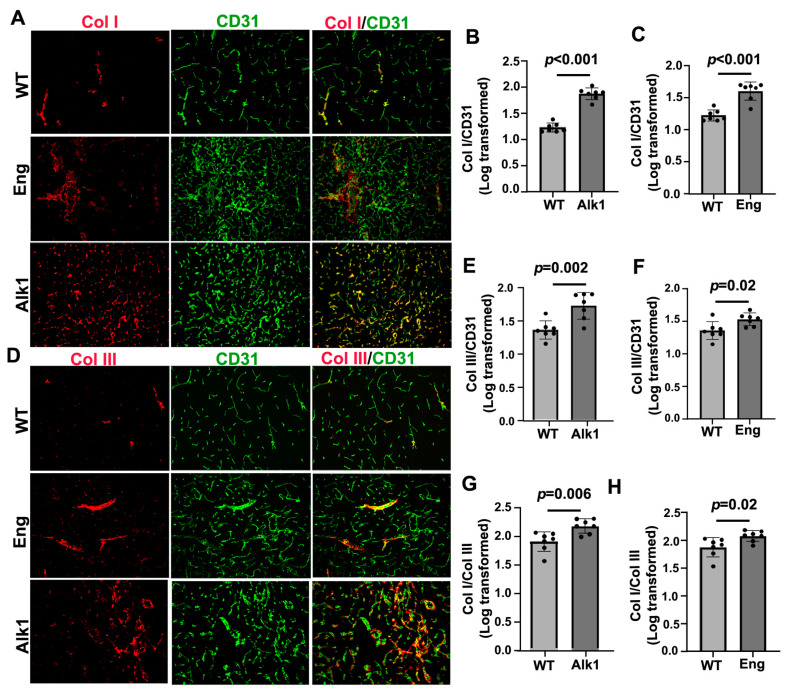
Col I, Col III, and Col I/Col III ratio were higher in mouse bAVMs than in the brain angiogenic region of control mice. (**A**,**D**) Representative images of Col I (red) and Col III (red) antibody-stained sections. ECs (green) were stained using an anti-CD31 antibody. Scale bar: 50 µm. (**B**,**E**) Quantifications of Col I and Col III levels in bAVMs in *Pdgfbi*creER;*Alk1*^f/f^ mice. (**G**) Quantification of Col I/Col III ratio in bAVMs in *Pdgfb*icreER;*Alk1*^f/f^ mice. (**C**,**F**) Quantifications of Col I and Col III levels in bAVMs in *Pdgfbi*creER;*Eng*^f/f^ mice. (**H**) Quantification of Col I/Col III ratio in bAVM in *Pdgfb*icreER;*Eng*^f/f^ mice. WT: corn-oil-treated controls; Alk1: *Pdgfb*icreER;*Alk1*^f/f^ mice; Eng: *Pdgfbi*creER;*Eng*^f/f^ mice. *n* = 7.

**Figure 4 cells-13-00092-f004:**
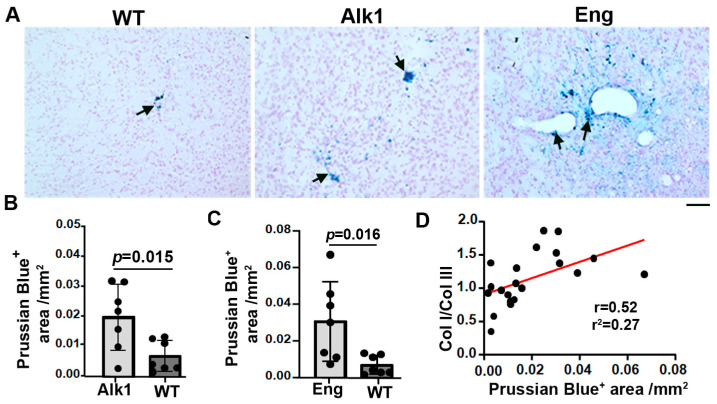
Microhemorrhage in mouse bAVMs. (**A**) Representative images of Prussian blue-stained sections. Iron-positive areas are stained blue (arrows). Scale bar: 50 µm. (**B**,**C**) Quantification of Prussian blue-positive areas. WT: corn-oil-treated controls; Alk1: *Pdgfb*icreER;*Alk1*^f/f^ mice; Eng: *Pdgfbi*creER;*Eng*^f/f^ mice. *n* = 7. (**D**) Correlations of Prussian blue-positive (Prussian Blue^+^) areas and Col I/Col III ratios.

**Figure 5 cells-13-00092-f005:**
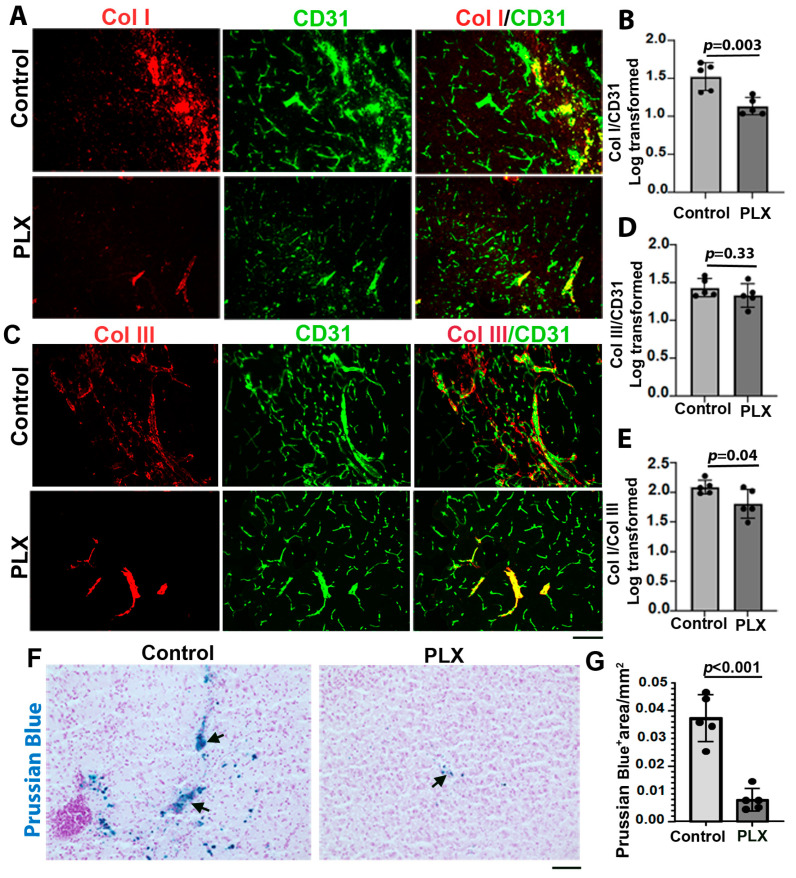
Transient depletion of CNS macrophage microglia reduced Col I/Col III ratio and microhemorrhage in bAVMs. (**A**,**C**) Representative images of Col I (red) and Col III (red) antibody-stained sections. ECs (green) were stained using an anti-CD31 antibody. Scale bar: 50 µm. (**B**,**D**) Quantifications of Col I and Col III levels in bAVMs of PLX5622 (PLX)-treated mice and placebo-chow-treated mice (control). (**E**) Quantification of Col I/Col III ratio in bAVMs of PLX5622 (PLX)-treated mice and placebo-chow-treated mice (control). *n* = 5. (**F**) Representative images of Prussian blue-stained sections. Arrows: positive staining. Scale bar: 50 µm. (**G**) Quantification of Prussian blue-positive (Prussian blue^+^) areas. *n* = 5.

**Figure 6 cells-13-00092-f006:**
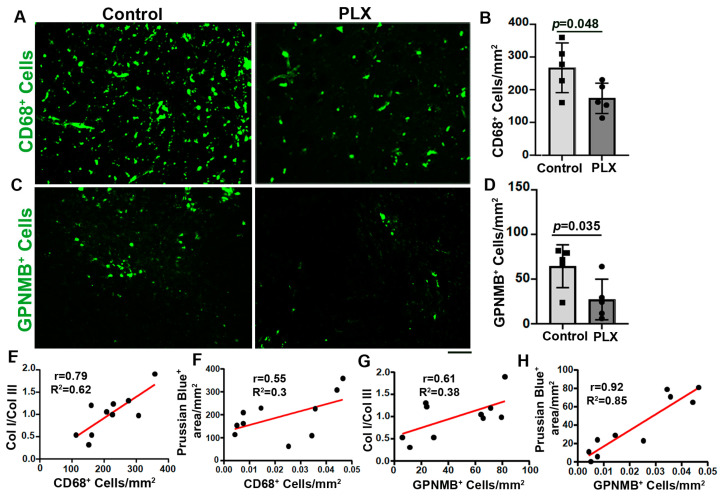
PLX5622 treatment reduced CD68^+^ cells and GPNMB^+^ cells in bAVMs. (**A**–**C**) Representative images of sections stained with antibodies specific to CD68 (**A**) or GPNMB (**C**). Scale bar: 50 µm. (**B**,**D**) Quantification of CD68^+^ cells (**B**) and GPNMB^+^ cells (**D**). *n* = 5. (**E**,**F**) Correlations of CD68^+^ cells with Col I/Col III ratio (**E**) and Prussian blue^+^ areas (**F**). (**G**,**H**) Correlations of GPNMB^+^ cells with Col I/Col III ratio (**G**) and Prussian blue^+^ areas (**H**). PLX: PLX5622-treated mice; control: placebo-chow-treated mice.

**Table 1 cells-13-00092-t001:** Upregulated pathways and expression of genes related to angiogenesis, inflammation, and collagen stability in *Eng*-deficient mouse bAVMs.

Pathway Name	Adj. *p*-Value
Angiogenesis	2.07 × 10^−17^
Vasculogenesis	9.6 × 10^−05^
Collagen binding signaling	0.0001
Acute inflammatory response	0.042
**Gene Name**	**Adj. *p*-Value**
CD248 *	1.8 × 10^−05^
Mpeg1 **	1.7 × 10^−05^
Icam2 **	0.0003
CD68 **	0.004
Csf1 **	0.002
Timp2	0.01
Vim *	0.019
CD34 *	0.02
Col1a1 ***	0.035
Col3a1 ***	0.032
Csf2rb **	0.049
Timp2 ****	0.01
Timp3 ****	0.04

Notes: O analysis was used to evaluate pathway changes and differential analysis was used to assess differential gene expression. * Fibroblast-specific genes. ** Inflammatory-related genes. *** Chains of Col I and Col III. **** Genes related to collagen degradation.

## Data Availability

The authors declare that all data supporting the findings of this study are available in the paper and its Appendix A.

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
