# Peer review of "Increased Collagen I/Collagen III Ratio Is Associated with Hemorrhage in Brain Arteriovenous Malformations in Human and Mouse"

_cells, 2024, doi:10.3390/cells13010092_

Round 1

Reviewer 1 Report

Comments and Suggestions for Authors

Journal: Cells (ISSN 2073-4409)

Manuscript ID: cells-2776631

Type: Article

Title: Increased Collagen I/ Collagen III ratio is associated with hemorrhage in brain arteriovenous malformations in human and mouse

Topic:Blood Physiology: Molecular Mechanisms of Vascular Wall Functioning, 2nd Volume

The article has been revised by the reviewer and the opinions and suggestions regarding the article are as follows;

1- Regarding the Abstract Section

 1-The experimental group of the hypothesis stated in the 'Background' section should be stated.

2- In the 'Methods' section, the sample size and characteristics of the samples should be stated in 'single or in 2 groups, if any', and if there is a control group, this should be stated. The reason for including the specified 'activin receptor-like kinase 1' or 'endoglin deletion' markers in the study and their relationship with the parameters to be examined should be stated.

3- If the demonstration of 'transient depletion of microglia' in the 'Results' section is within the scope of the aims of the study, this should be stated in the background/aim section.

2- Regarding the Materials and Methods Section

1. There is a possibility that tissue damage/sclerosis that may be seen in individuals diagnosed with 'temporal lobe epilepsy' in the control group included in the Ethics statement section may affect the results of the study in the form of false positivity or negativity. This problem should be clarified or the control group sample should be changed.

2- The study includes an animal experiment group and at least one of the authors must have a certificate to conduct experiments on animal subjects. In this respect, the certified author who contributed to the study conducted in this sample group should be stated.

3- In the '2.2. 'Human Specimens' section, it should be stated whether the tissues obtained from the sample group were studied at the same time or, if not simultaneously, what precautions were taken, considering that the degeneration that may occur in the tissue may affect the results of the study. In addition, the criteria for inclusion/exclusion in the study, including the demographic information of individuals in the sample group, should be clearly explained.

4- In the '2.3. Animals' section, the criteria for including or not including samples in the study should be clearly explained. In addition, evidence and reference information must be provided that the subjects used for induction will make changes in the relevant parameter (i.e.1) PdgfbicreER;Engf/f.; 2) PdgfbicreER;Alk1f/f and 3) Alk1f/f. The PdgfbicreER;Engf/f and PdgfbicreER;Alk1f/f).

5- In the ‘2.4. Brain AVM model induction and viral vector delivery in mice' section, the sample number and method rationale for each method used in the 'Administration of CSF1R inhibitor (PLX5622)' sections should be clearly stated, including the concept of time.

6- In the '2.6.'Immunofluorescence staining of human and mouse sections' section, the method(s) used for the collection, storage and working conditions of the material should be referenced.

7- In the ‘2.8. RNA-Seq analysis' section, the 'brain angiogenic regions (controls)' specified should be topographically localized and detailed, which part is examined with which imaging method.

3- Regarding the Statistics Section

1- This section should be expanded by reviewing the dependent and independent variables and including Power Analysis information about the method used.

4-Regarding the Results Section

1-In the study, high COL I and COL III levels were found in both groups, but this elevation did not lead to statistical significance between groups with bAVMS. Statistical significance was reached when compared with the control group. These reasons '3.1. The title stating 'Human ruptured bAVMs have higher COL I and COL III levels ..' and the inference in the relevant paragraph should be corrected.

2- In the '3.3. Transient depletion of Central nervous system (CNS) macrophages/microglia reduced Col I/Col III ratios and microhemorrhage in bAVM' section, it should be detailed what inflammation is meant in the sentence '...inflammation by transient depletion of CNS macrophages/microglia...' and should be referenced.

3- In the '3.4. Fibroblast and inflammatory-related genes are upregulated in the mouse bAVMs' section, the method used for RNA sequencing should be detailed and referenced. Additionally, this part should be explained in the material/method section.

4-The study was conducted on two different species, and the similarities and differences in the relevant parameters should be detailed in the results obtained.

5- Regarding the Discussion Section

1-The ‘Discussion’ section should be supported by current literature, and the techniques included throughout the article should be referenced, taking into account any studies, including Pilot experiments, that have been done on this subject so far.

The study can be re-evaluated for publishability after necessary corrections in the light of current recommendations.

Author Response

We thank this reviewer for his/her through review and great suggestions, and for giving us the opportunity for getting our manuscript re-evaluated after revision. We have addressed the reviewer’s comments and revised the manuscript based on our best understanding of the questions.

The article has been revised by the reviewer and the opinions and suggestions regarding the article are as follows.

1-Regarding the Abstract Section

1-The experimental group of the hypothesis stated in the 'Background' section should be stated.

Response: We have added human and mouse bAVM in the hypothesis statement. Due to the word limitation (200 words) for the abstract, we could not include all experiment groups in the background section in the Abstract.

2-In the 'Methods' section, the sample size and characteristics of the samples should be stated in 'single or in 2 groups, if any', and if there is a control group, this should be stated. The reason for including the specified 'activin receptor-like kinase 1' or 'endoglin deletion' markers in the study and their relationship with the parameters to be examined should be stated.

Response: We have added sample sizes, groups, and controls to the Methods section. The following sentence has been added to explain the reason of including of mouse models.

“Mouse bAVMs were used to test the relationships of Col I/Col III ratio with hemorrhage and if transiently depletion of microglia reduces Col I/Col III ratio and hemorrhage.”

3- If the demonstration of 'transient depletion of microglia' in the 'Results' section is within the scope of the aims of the study, this should be stated in the background/aim section.

Response: We have added 'transient depletion of microglia' to the background and hypotheses in the abstract.

2- Regarding the Materials and Methods Section

1-There is a possibility that tissue damage/sclerosis that may be seen in individuals diagnosed with 'temporal lobe epilepsy' in the control group included in the Ethics statement section may affect the results of the study in the form of false positivity or negativity. This problem should be clarified, or the control group sample should be changed.

Response:  The tissues used in this study were selected from tissue bank of the clinical group in the Center for Cerebrovascular Research directed by co-author, Dr. Helen Kim. Please also see answer to comment 3 below. Samples that have bAVM tissues or temporal lobe tissues without damage/sclerosis were banked and used in this study. We have added this information in Materials and Methods, 2.1. Ethics statement.

2- The study includes an animal experiment group and at least one of the authors must have a certificate to conduct experiments on animal subjects. In this respect, the certified author who contributed to the study conducted in this sample group should be stated.

Response: Authors: Zahra Shabani, Xiaonan Zhu, Chaoliang Tang, Li Ma, Alka Yadav, Rich Liang, Kelly Press, Calvin Wang, Abinav Sekhar, and Hua Su have been authorized by the Institutional Animal Care and Use Committee (IACUC) of the University of California, San Francisco to perform experiments on mouse. We have added this information to the Section of Institutional Review Board Statement and in Section 2.1. Ethics statement

3- In the '2.2. 'Human Specimens' section, it should be stated whether the tissues obtained from the sample group were studied at the same time or, if not simultaneously, what precautions were taken, considering that the degeneration that may occur in the tissue may affect the results of the study. In addition, the criteria for inclusion/exclusion in the study, including the demographic information of individuals in the sample group, should be clearly explained.

Response: We have added following statement in Materials and Methods, 2.1. Ethics statement to clarify the origination of the samples and the criteria for inclusion/exclusion in this study.

“In this study, we have taken the advantage of a large cohort of patients with bAVM in place as part of the UCSF Brain AVM Project, led by co-author, Dr. Helen Kim (R01 NS034949). Dr. Kim’s team discussed with all patients for consent to participate in our research studies, which allowed for tissue to be harvested, banked, and analyzed for medical research. For this study, we used de-identified tissue specimens from 12 (7 unruptured and 5 ruptured) bAVM subjects undergoing surgery, with appropriate consents. Nonvascular lesion controls, temporal lobe specimens were selected from subjects that matched to bAVM cases using the demographic distribution from enrolled cases at UCSF.”

We have also clarified in section 2.2. Human Specimens that all samples were analyzed at the same time. The demographic information of individuals in the sample group is shown in newly added Supplementary Table 1.

4- In the '2.3. Animals' section, the criteria for including or not including samples in the study should be clearly explained. In addition, evidence and reference information must be provided that the subjects used for induction will make changes in the relevant parameter (i.e.1) PdgfbicreER;Engf/f.; 2) PdgfbicreER;Alk1f/f and 3) Alk1f/f. The PdgfbicreER;Engf/f and PdgfbicreER;Alk1f/f).

Response: Following inclusion/exclusion criteria have been added to Section 2.3. Animals.

“The inclusion criteria were: (1) 8- to 10-weeks of age; (2) no wound infection and neuronal deficits after receiving the model induction procedures (please see Section 2.5). The exclusion criteria were: (1) mice that were younger than 8 weeks or older than 10 weeks or pregnant; (2) have wound infection or neuronal deficits after receiving the model induction procedures (please see Section 2.5); (3) brain samples were damaged during sample collection, section or staining (no sample damage had occurred during this study).”

We do not fully understand the meaning of the comment: “In addition, evidence and reference information must be provided that the subjects used for induction will make changes in the relevant parameter (i.e.1) PdgfbicreER;Engf/f.; 2) PdgfbicreER;Alk1f/f and 3) Alk1f/f. The PdgfbicreER;Engf/f and PdgfbicreER;Alk1f/f).” We hope the following changes have answered these comments.

The references for PdgfbiCreER[1], Alk1f/f [2] and Engf/f [3] transgenes are cited in Section 2.3, Animals. The PdgfbicreER;Engf/f and PdgfbicreER;Alk1f/f have pdgfb promoter driven estrogen inducible cre recombinase expression in the endothelial cells (ECs) [1] and have Eng gene exons 5-6 floxed [3] or Alk1 gene exons 4-6 floxed [2]. Alk1f/f mice have Alk1 gene exons 4-6 floxed [2]. Using these mouse lines, we can test the effect of two AVM causative genes, Eng and Alk1, and two gene deletion sides, in ECs or in brain locally.

5- In the ‘2.4. Brain AVM model induction and viral vector delivery in mice' section, the sample number and method rationale for each method used in the 'Administration of CSF1R inhibitor (PLX5622)' sections should be clearly stated, including the concept of time.

Response: The sample were added to section 2.3, Animals, Section 2.4, Brain AVM model induction and viral vector delivery in mice and Section 2.5. Administration of CSF1R inhibitor (PLX5622). The rationales for each method used to induce bAVM models have also been added to Section 2.4.

The PLX5622 was incorporated in chow and orally administered for 7 consecutive days starting at 8 weeks after bAVM model induction in Alk1f/f mice (n=5) based on the instructions provided by Plexikon Biotech Company. We have clarified the description in Section 2.5. Administration of CSF1R inhibitor (PLX5622).

6- In the '2.6.'Immunofluorescence staining of human and mouse sections' section, the method(s) used for the collection, storage and working conditions of the material should be referenced.

Response: References were added to 2.6. Immunofluorescence staining of human and mouse sections.

7- In the ‘2.8. RNA-Seq analysis' section, the 'brain angiogenic regions (controls)' specified should be topographically localized and detailed, which part is examined with which imaging method.

Response: Brain AVM lesions and brain angiogenic regions (controls) were identified by needle injection holes on the brain surface and the deep of needle tips (3 mm beneath brain surface, please see Section 2.4. Brain AVM model induction and viral vector delivery in mice). One mm3 tissue around the needle tip was collected from each brain for RNA isolation. We have added this information to Section 2.8. RNA-Seq analysis.

3- Regarding the Statistics Section

This section should be expanded by reviewing the dependent and independent variables and including Power Analysis information about the method used.

Response: We have added the dependent and independent variables to the statistics Section. The sample sizes were determined using power analysis based on pilot studies. For example: The differences in COL I levels between ruptured and control samples was quite pronounced (2.49±0.12 vs.1.87±0.03). Given those effect sizes and standard deviations, we would require only n=3 per group to find a difference with 80% power and an alpha of 0.05 when using a two-sample t-test. We used n=6 for the control group and n=5 for the ruptured group, so we were sufficiently powered to detect a difference.

4-Regarding the Results Section

1-In the study, high COL I and COL III levels were found in both groups, but this elevation did not lead to statistical significance between groups with bAVMS. Statistical significance was reached when compared with the control group. These reasons '3.1. The title stating 'Human ruptured bAVMs have higher COL I and COL III levels ..' and the inference in the relevant paragraph should be corrected.

Response: We have changed the title of Section 3.1 to” Human ruptured bAVMs have higher COL I/ COL III ratio than unruptured bAVMs” and corrected the inference in the relevant paragraphs.

2- In the '3.3. Transient depletion of Central nervous system (CNS) macrophages/microglia reduced Col I/Col III ratios and microhemorrhage in bAVM' section, it should be detailed what inflammation is meant in the sentence '...inflammation by transient depletion of CNS macrophages/microglia...' and should be referenced.

Response: We have revised this sentence to “To test if reduction of inflammation in mouse bAVMs, such as increased inflammatory cells[4], by transient depletion of CNS macrophages/microglia influences Col I and Col III levels, and Col I/Col III ratios in bAVMs, Alk1f/f mice were treated with PLX5622 8 weeks after bAVM model induction for 7 days.”

3- In the '3.4. Fibroblast and inflammatory-related genes are upregulated in the mouse bAVMs' section, the method used for RNA sequencing should be detailed and referenced. Additionally, this part should be explained in the material/method section.

Response: The detailed information for RNA-Seq method and analysis are now provided in Supplementary Material 1 which is cited in Material and Methods, Section 2.8. RNA-Seq analysis. 

4-The study was conducted on two different species, and the similarities and differences in the relevant parameters should be detailed in the results obtained.

Response: The similarities in the relevant parameters between human and mouse bAVMs were summarized in the first paragraph of Discussion Section. We have not detected differences between human and mouse bAVMs in the parameters used in this study. 

5- Regarding the Discussion Section

1-The ‘Discussion’ section should be supported by current literature, and the techniques included throughout the article should be referenced, taking into account any studies, including Pilot experiments, that have been done on this subject so far.

Response: We have added references to Material and Methods sections and many recent papers and data to the Discussion Section. For example: these sentences have been added to the second paragraph of Discussion Section:

“Fu et al. showed that vascular fibrosis was a common pathological manifestation in bAVMs with or without microhemorrhage. However, collagen deposition was notably higher in bAVMs with microhemorrhage than those without [5]. Their data confirmed the correlation of collagen with bAVM microhemorrhage. However, they did not distinguish different types of collagens in their study. The contribution of COL I and COL III imbalance to bAVM hemorrhage has also been reported by Neu et al. [6].”

References

  1. Payne, S., S. De Val, and A. Neal, Endothelial-Specific Cre Mouse Models. Arteriosclerosis, Thrombosis, and Vascular Biology, 2018. 38(11): p. 2550-2561.
  2. Park, S.O., et al., ALK5- and TGFBR2-independent role of ALK1 in the pathogenesis of hereditary hemorrhagic telangiectasia type 2 (HHT2). Blood, 2008. 111(2): p. 633-42.
  3. Allinson, K.R., et al., Generation of a floxed allele of the mouse Endoglin gene. Genesis, 2007. 45(6): p. 391-395.
  4. Zhang, R., et al., Persistent infiltration and pro-inflammatory differentiation of monocytes cause unresolved inflammation in brain arteriovenous malformation. Angiogenesis, 2016. 19(4): p. 451-461.
  5. Fu, W., et al., Mesenchymal behavior of the endothelium promoted by SMAD6 downregulation is associated with brain arteriovenous malformation microhemorrhage. Stroke, 2020. 51(7): p. 2197-2207.
  6. Niu, H., et al., Relationships between hemorrhage, angioarchitectural factors and collagen of arteriovenous malformations. Neuroscience bulletin, 2012. 28: p. 595-605.

Reviewer 2 Report

Comments and Suggestions for Authors

The manuscript titled "Increased Collagen I/ Collagen III Ratio is Associated with 2 Hemorrhage in Brain Arteriovenous Malformations in Human 3 and Mouse" establishes as its main hypothesis that COLI/COL III ratio is increased in the brain arteriovenous malformations (bAVMs), which is associated with bAVM hemorrhage. 

They have animal models and human samples. However, according to the hypothesis and introduction, it is not clear how the use of the animal models will clarify the role of COLI/COLII and bAVMs. The wording of the objective is unclear, and this is combined with the lack of relevant information in the introduction. Therefore, the justification and relevance of each experiment is not entirely clear.
Relevant information such as that related to RNAseq analysis is not presented in the brief or is presented as supplementary information. 

In general, it is important to restructure the writing and presentation of results to make the information and relevance of the study clearer.

Author Response

We thank this reviewer for his/her comments and suggestions. We have addressed the reviewer’s comments and revised the manuscript accordingly.

They have animal models and human samples. However, according to the hypothesis and introduction, it is not clear how the use of the animal models will clarify the role of COLI/COLII and bAVMs. The wording of the objective is unclear, and this is combined with the lack of relevant information in the introduction. Therefore, the justification and relevance of each experiment is not entirely clear.

Response: Since human samples can only be use for observation study, we included mouse bAVM models to test possible mechanism that underlying the observed correlation of COL I/COL III ratio with hemorrhage in human bAVMs and to test potential interaction that may be used to reduce bAVM hemorrhage. We have added this information to the last paragraph of Introduction Section.

Relevant information such as that related to RNAseq analysis is not presented in the brief or is presented as supplementary information. 

Response: The detailed information for RNA-Seq method and analysis are now provided in Supplementary Material 1 which is cited in Material and Methods, Section 2.8. RNA-Seq analysis. 

In general, it is important to restructure the writing and presentation of results to make the information and relevance of the study clearer.

Response: We have revised the manuscript extensively and hope the information is clear now. 

Reviewer 3 Report

Comments and Suggestions for Authors

This study deals with Collagen I, Collagen III, and Collagen I/ Collagen III ratio with brain arteriovenous malformations in vivo.

You have submitted quite a fascinating manuscript.

I did not fully understand Collagen I/ Collagen III ratio, which was the subject of this manuscript.

I thought that the effects of Collagen I and Collagen III on AVM rupture were shown to be sufficiently understandable. I thought that since both Collagen I and Collagen III were increasing, there was little merit in taking the ratio. I didn’t think enough consideration had been made to overturn my hypothesis.

What is the main question addressed by the research?

Author Response

We thank this reviewer for his/her positive comments. We have addressed his/her questions in the revised manuscript.

Comments and Suggestions for Authors

This study deals with Collagen I, Collagen III, and Collagen I/ Collagen III ratio with brain arteriovenous malformations in vivo.

You have submitted quite a fascinating manuscript.

Response: We thank this reviewer for his/her positive comment.

I did not fully understand Collagen I/ Collagen III ratio, which was the subject of this manuscript.

I thought that the effects of Collagen I and Collagen III on AVM rupture were shown to be sufficiently understandable. I thought that since both Collagen I and Collagen III were increasing, there was little merit in taking the ratio. I didn’t think enough consideration had been made to overturn my hypothesis.

Response: Since COL I provides resistance to stretch, whereas COL III forms an elastic network that offers resilience of the vessel wall, increase COL I/COL III ratio will increase vessel stiffness and render the vessel less resistant to the changed of flow and pressure. Therefore, we also analyzed the COL I/COL III ratio to determine if there is any correlation between COL I/COL III ratio and bAVM hemorrhage. We have added this information to the last paragraph of the Introduction Section.

Round 2

Reviewer 1 Report

Comments and Suggestions for Authors

The revision changes suggested in the article have been made within the framework of the necessary corrections.

Reviewer 3 Report

Comments and Suggestions for Authors

I think that the revised parts have made it easier to understand and the main theme has been clarified.